# Microparticle Phosphatidylserine Mediates Coagulation: Involvement in Tumor Progression and Metastasis

**DOI:** 10.3390/cancers15071957

**Published:** 2023-03-24

**Authors:** Haijiao Jing, Xiaoming Wu, Mengqi Xiang, Chengyue Wang, Valerie A. Novakovic, Jialan Shi

**Affiliations:** 1Department of Hematology, The First Hospital, Harbin Medical University, Harbin 150001, China; 2Department of Research, VA Boston Healthcare System, Harvard Medical School, Boston, MA 02132, USA; 3Department of Medical Oncology, Dana-Farber Cancer Institute, Harvard Medical School, Boston, MA 02132, USA

**Keywords:** phosphatidylserine, microparticles, coagulation cascade, tumor progression, treatment strategies

## Abstract

**Simple Summary:**

Microparticles (MPs) play a key role in intercellular communication and mediate many features of cancers by delivering their various biomolecular cargos (including phospholipids). Increased release of phosphatidylserine-rich MPs from cancer and blood cells promotes tumor development and progression. Cancer cell signaling contributes to the formation of hypercoagulability, which in turn facilitates tumor progression. This review links coagulation activation with the process of metastatic tumor spread, including angiogenesis and matrix degradation. Therefore, anticoagulation can not only reduce thrombus formation but also slow tumor progression, which is of great significance for patient treatment. We provide promising cancer treatment strategies targeting the phosphatidylserine-mediated coagulation cascade and fill current knowledge gaps.

**Abstract:**

Tumor progression and cancer metastasis has been linked to the release of microparticles (MPs), which are shed upon cell activation or apoptosis and display parental cell antigens, phospholipids such as phosphatidylserine (PS), and nucleic acids on their external surfaces. In this review, we highlight the biogenesis of MPs as well as the pathophysiological processes of PS externalization and its involvement in coagulation activation. We review the available evidence, suggesting that coagulation factors (mainly tissue factor, thrombin, and fibrin) assist in multiple steps of tumor dissemination, including epithelial–mesenchymal transition, extracellular matrix remodeling, immune escape, and tumor angiogenesis to support the formation of the pre-metastatic niche. Platelets are not just bystander cells in circulation but are functional players in primary tumor growth and metastasis. Tumor-induced platelet aggregation protects circulating tumor cells (CTCs) from the blood flow shear forces and immune cell attack while also promoting the binding of CTCs to endothelial cells and extravasation, which activates tumor invasion and sustains metastasis. Finally, in terms of therapy, lactadherin can inhibit coagulation by competing effectively with coagulation factors for PS binding sites and may similarly delay tumor progression. Furthermore, we also investigate the therapeutic potential of coagulation factor inhibitors within the context of cancer treatment. The development of multiple therapies targeting platelet activation and platelet–tumor cell interactions may not only reduce the lethal consequences of thrombosis but also impede tumor growth and spread.

## 1. Introduction

Extracellular vesicles (EVs) play a prominent role in promoting cancer development and progression [1,2]. Microparticles (MPs, 100–1000 nm) are a heterogeneous population of EVs that are formed from the direct budding of the plasma membrane [3,4]. Accumulating evidence suggests that enhanced secretion of MPs by cancer cells and PS externalization on their surface are associated with tumor proliferation and progression [5,6]. Sustained delivery of PS can promote malignant ascites and subcutaneous tumor growth. PS externalization on apoptotic tumor cells promotes the formation of immunosuppressive microenvironments by inducing polarization and accumulation of macrophages, and its related pathways can serve as potential targets for tumor therapy [7]. Additionally, PS exposure on the tumor cell surface contributes to evading immune surveillance and thus resulting in tumor spread. The coagulation system is currently recognized as playing a major role in the regulation and progression of malignancies. Indeed, several components of the clotting system influence many known processes of tumor progression, including tumor invasion and metastasis [8,9]. Moreover, the coagulopathy score can be used to predict the postoperative recurrence of gastric cancer [10]. PS-medicated coagulation has been described in malignant tumors, including breast cancer, multiple myeloma, and acute promyelocytic leukemia. PS+ breast cancer cell-derived MPs stimulate platelet activation and apoptosis, enhancing their capacity to promote fibrin formation and increasing the incidence of microvascular obstruction in vital organs [11]. Interestingly, PS exposure to MPs is shown to contribute to hypercoagulable states, and higher rates of thromboembolic events are related to oral immunomodulatory drug use, especially thalidomide [12]. The exposure of PS on apoptotic acute promyelocytic leukemia cells supports increased generation of intrinsic FXase and prothrombinase complexes [13]. However, whether PS-mediated coagulation contributes to tumor progression and cancer metastasis remains unclear. Cancer metastasis is the main cause of death worldwide, and the dismal survival status following metastasis highlights the urgency of better understanding the molecular mechanisms underlying tumor progression to develop effective treatment strategies. More mechanisms regarding tumor metastasis are being investigated to prolong survival in cancer patients. 

## 2. MPs Involved in Tumor Progression

MPs can be directly released by cancer cells or from other types of cells (platelets, neutrophils, etc.) induced by tumor cell signaling or damage. In blood circulation, MPs originate from vascular endothelial cells (ECs) and blood cells, the most prominent of which are platelets [14,15,16]. MPs are produced from cells that are undergoing activation or apoptosis. Furthermore, MPs can also be released from resting-state platelets through αIIβ3 integrin-mediated actin cytoskeleton destabilization independent of calpain activation [17]. Cancer cells shed significantly more MPs compared to normal cells [18]. Tumor cells proliferate very rapidly, leading to inadequate blood supply followed by cell apoptosis, resulting in the production of tumor-derived MPs (T-MPs). Additionally, changes in the microenvironment can also stimulate or cause the death of tumor cells and generate MPs, such as hypoxia and various cell damage-associated and apoptotic signals [19,20]. Typically, tumor cells express Toll-like receptor 4 (TLR4), and the activation of TLR4 signaling pathways results in cytoskeleton modifications, causing T-MPs to shed from cell membrane surfaces [21,22]. Studies have revealed that high levels of MPs correlate strongly with inflammation-associated cytokine secretion in cancer cells [23,24]. In practice, tumor cell death induced by radiation, chemotherapy, and targeted therapies inevitably leads to elevated levels of T-MPs [25]. Therefore, the generation of T-MPs is a quite common phenomenon in the tumor microenvironment (TME). Once generated, T-MPs are able to enter the circulation through leaky tumor vasculature.

### 2.1. The Effect of MPs on Tumor Progression

MPs that are induced by environmental cues (damage, hypoxia, or apoptosis) participate in cancer initiation, progression, and metastasis [26,27]. Cancer-associated MPs were first described in the plasma of patients with Hodgkin’s disease. Levels of circulating MPs are consistently increased in various types of cancer, such as hematologic malignancies, breast cancer, and colorectal cancer [28,29]. At the cellular and protein level, the number of circulating MPs is highly related to pro-angiogenic factors [30]. Ko et al. showed that rats treated with lung cancer patient-derived circulating MVs exhibited a significantly higher microvascular count, more CXCR4+ (C-X-C motif chemokine receptor 4) and VEGF+ (vascular endothelial growth factor) cells, and accelerated growth of pulmonary-metastatic hepatocellular carcinoma [31]. As the disease progresses, the number of circulating platelet-derived MPs (PMPs) increases further. For example, in gastric cancer, the levels of PMPs are significantly higher in stage IV than in stage I or II/III tumors and correlate with plasma vascular endothelial growth factor (VEGF) and IL-6 levels [32]. Interestingly, PMP levels have the highest diagnostic accuracy and can predict distant metastasis [33]. Endothelium-derived MPs carry transcripts, induce angiogenic activity primarily in quiescent ECs, promote endothelial cell proliferation and the formation of capillary-like structures, and prevent apoptosis [26]. Elevated levels of circulating lymphocyte-derived MPs (LMPs) are related to the disease progression of advanced non-small cell lung cancer (NSCLC) [34]. T-MPs have been shown to affect distant cellular niches, establish a favorable microenvironment, and support the growth of spreading cancer cells [30]. Additionally, T-MPs are absorbed by immune cells (e.g., macrophages) and result in the inhibition and alteration of the anti-tumor immune response [25,35]. MP signaling enhances the immunosuppressive properties of tumor cells and promotes the evasion of immune surveillance and cancer metastasis. 

### 2.2. Biogenesis of MPs

To date, the full mechanism of MP formation is not yet fully understood. We, therefore, summarize the general process of MP biogenesis. The phospholipid composition and distribution in the cell membrane are highly specific: the outer membrane consists primarily of phosphatidylcholine and sphingomyelin (SM), while PS and phosphatidylethanolamine reside predominantly on the inner membrane [36]. Normally, enzymes, such as flippase and scramblase, maintain dynamic asymmetric homeostasis by selectively transporting phospholipids (including phosphatidylcholine) across the membrane [35]. However, activation and apoptosis cause increased cytosolic Ca^2+^ concentration, disrupting membrane asymmetry and causing rearrangement of the actin cytoskeleton, thereby resulting in MP release (Figure 1). First, Ca^2+^ released from the endoplasmic reticulum activates several Ca^2+^-dependent enzymes, such as flippase and scramblase, which transport PS to the cell surface and affect the phospholipid structure of the cell membranes [37]. Notably, PS exposure does not occur throughout the population of MPs, suggesting that other mechanisms are also implicated in MP budding, potentially involving other lipids and their structural domains. Cholesterol is a lipid component that is abundant in MPs, and its pharmacological consumption can impair MP formation from activated neutrophils [38]. It has been reported that the spontaneous bending of transmembrane proteins also leads to membrane lipid asymmetry, which further increases local membrane curvature [35,39]. Cytoskeletal elements and their regulators are also essential for MP formation. Contraction of actin-myosin fibers also promotes MP fission and release. The process of MP release requires ATP-dependent actomyosin contractile machinery, which constricts the neck of MPs [40]. This contractile mechanism facilitates the sliding of the membrane to the microvillus tip, where membrane vesiculation induces the formation of MPs [41,42]. 

### 2.3. Factors and Mechanisms Influencing MP Biogenesis

Numerous studies have suggested the cytoskeleton exerts essential roles in diverse biological activities, including MP release and uptake [43,44]. Actin dynamics are regulated by the ADP-ribosylation factor (ARF) and the Rho family, which act through the actin cytoskeleton to facilitate MP fission from cancer cells. ARF is a member of the small GTP-binding protein subfamily consisting of ARF1-6. Activation of RhoA and RhoC is modulated by ARF1, which causes the phosphorylation of the myosin light chain (MLC) and the contraction of actomyosin, thereby promoting MP release [45]. RhoA and its downstream effector, Rho-associated protein kinase (ROCK), have been shown to activate LIM kinase (LIMK), which can phosphorylate cofilin and inhibit its actin-severing activity [46]. The formation of MPs in cancer cells depends on the LIMK-cofilin signaling pathway activated by ROCK. Additionally, RhoA/ROCK can also activate extracellular signal-regulated kinase (ERK) and then inhibit myosin light chain phosphatase (MLCP), which can inactivate MLC, further facilitating MP secretion [47]. ARF6 regulates MLC activity and MP shedding via multiple pathways. ARF6 activation by phospholipase D (PLD) can recruit ERK to activate MLC kinase, thus facilitating the release of MPs. The inhibition of ARF6 activation is accompanied by PKC-mediated MLC phosphorylation, which reduces MLC activity and prevents MP shedding [40]. The activation of Rab22a increases MP generation in breast cancer, which is associated with ARF6-regulated MP transport. Interestingly, Rab22a knockout reduces hypoxia-induced MP production while only marginally affecting MP biogenesis under non-hypoxic conditions [48]. Additionally, peptidyl arginine deiminase (PAD) influences cytoskeletal rearrangement by mediating the deamination of actin. In prostate cancer cells (PC3), increased expression levels of calcium-dependent PAD2 and PAD4 and deamination of the actin cytoskeleton further promote the shedding of MPs [49]. In multiple cancer cells, v-H-Ras expression causes ERK-dependent chromosome segregation 1-like (CSE1L) phosphorylation and MP biogenesis. The overexpression of CSE1L causes melanoma cells (B16F10) to generate MPs, and the knockdown of CSE1L reduces v-H-Ras-induced MP formation, matrix metalloproteinase 2 (MMP2) and MMP9 secretion as well as metastasis [50]. Additionally, miRNA is also involved in the process of MP generation as miR-200a is shown to suppress MP release in hepatocellular carcinoma cells. It has been found that miR-200a targets gelsolin and changes the cytoskeleton to modulate MP secretion [51]. 

## 3. PS Externalization on MPs Involved in Tumor Coagulation

MP-mediated material transfer to adjacent or remote cells has been shown to affect many stages of cancer progression [35]. MPs can encapsulate and transfer molecules between cells, including nucleic acids (DNA, mRNA, microRNA, etc.), proteins (receptors, enzymes, transcription factors), and lipids, thereby facilitating intercellular communication in most physiological and pathological situations [52]. Phospholipids, primarily PS, phosphatidylcholine, and phosphatidylethanolamine, are major components of MP plasma membranes. PS exposure on the outer leaflet of the plasma membrane results from various stimuli [53]. Matsumura et al. [54] reported cancer cells and their derived MPs having variable degrees of PS externalization. Compared with normal cells, cancer cells are less able to maintain PS asymmetry, resulting in PS exposure on the cell surface [55]. PS externalization is of great significance for tumor cell proliferation and cancer-related symptoms. Increased PS is associated with the invasiveness of cancer cells, while PS on MPs is related to neovascularization [56]. Additionally, PS+ EVs in body fluids are considered important diagnostic tools for tumor progression [57,58]. 

### 3.1. The Mechanisms of PS Externalization 

PS distribution between the inner and outer leaflets of the plasma membrane is an important physiological signal. Increased PS externalization on cancer cells and their released MPs are generally coordinated by flippase and scramblase. Type IV P-type ATPases (P4-ATPases), known as flippases, transport specific lipid substrates (PS) from the exofacial leaflet to the cytosolic leaflet [59,60]. In humans, there are fourteen intramembranous ATPases distributed in the different tissues and membrane compartments, among which ATP8A1, ATP11A, and ATP11C are found to catalyze phospholipid translocation [61]. When the cytosolic Ca^2+^ concentration is elevated owing to various endogenous and exogenous stimuli, ATP11A and ATP11C are inactivated, thereby promoting PS exposure on the cell surface. Furthermore, CDC50 family proteins (especially CDC50A) are necessary for normal flippase activity, and the deficiency of CDC50A leads to the inhibition of flippase function and promotes the process of PS exposure [62]. In contrast, flippase inactivation by caspase cleavage is insufficient to expose PS, as the internal and external transmembrane exchange of any phospholipids does not spontaneously occur owing to a high energy barrier [63]. The movement of PS from the inner to the outer leaflet of the plasma membrane requires the action of another enzyme, the so-called scramblase. Scramblase mediates a non-specific, bidirectional, and ATP-independent phospholipid movement. Xk-related protein 8 (Xkr8) and transmembrane protein 16F (TMEM16) are identified as two kinds of scramblases for PS transport. Xkr8 has been found to increase PS externalization in response to apoptotic stimuli, including oxidative stress and DNA degradation. The molecular mechanisms of Xkr8 activation involve caspase-dependent as well as phosphorylated regulatory signaling pathways [64]. Caspase-dependent Xkr8 activation and P4-ATPase inactivation collectively lead to persistent PS exposure in apoptotic cells (Figure 1). TMEM16-mediated Ca^2+^ activation catalyzes PS exposure on activated platelets. The TMEM16/anoctamin family of proteins is composed of ten transmembrane paralogs, ranging from TMEM16A- H to TMEM16 J-K. In the process of TF activation controlled by caspase-11- and gasdermin D- dependent pathways, PS exposure is mediated by TMEM16F (anoctamin 6, Figure 1) [65].

### 3.2. PS Involved in Tumor Coagulation

Up to 20% of cancer patients develop vascular thromboembolism, consisting of pulmonary embolism (PE) and deep vein thrombosis (DVT). The risk of thrombosis is particularly high in patients with hematologic malignancies, pancreatic cancer, and gastric cancer. Cancer is related to a four- to seven-times increased risk of venous thromboembolism (VTE) [66]. Patients undergoing chemotherapy have a 4.8-fold increased risk of VTE within 3 months. In contrast, radiotherapy does not increase the risk of VTE [67]. VTE, especially PE and DVT, have become the second leading cause of death in cancer patients. Even in the absence of thrombosis, most cancer patients typically present with abnormal laboratory coagulation parameters, and these alterations reveal different degrees of coagulation activation, suggesting a constant subclinical hypercoagulable state. These changes include shortened activated partial thromboplastin time, increased platelets, elevated levels of circulating coagulation factors (fibrinogen, Factor V (FV), FVIII, FIX, and FX) as well as increased fibrin degradation products [68]. Cancer-driving events specifically contribute to hypercoagulability, the mechanisms of which include increased activation of procoagulant factors, impaired fibrinolysis, secretion of proinflammatory cytokines as well as adhesive effects between tumor cells, ECs, and blood cells [69]. Recently, studies have shown that PS exposure to cancer cells and their released MPs exert primary roles in blood clotting. The mechanisms by which PS participates in coagulation are as follows.

One of the most prominent mechanisms in many cancers, particularly in pancreatic cancer, is enhanced activation of the TF coagulation pathway by upregulating TF on the tumor cell surface and releasing TF-bearing MPs (TF-MPs) [69]. The TF on MPs binds to circulating activated factor VII (FVIIa) to form complexes and then activates FIX and FX. FXa, in turn, activates more FVII, forming a positive feedback loop promoting coagulation. PS can convert cryptic TF to active TF, dramatically amplifying its role as the primary initiator of coagulation. Both TF and PS aggregate on the surface of circulating T-MPs and can trigger venous thrombosis in the absence of vascular injury [70]. In prostate cancer, the polyphosphate FXII pathway is considered a major driver of thrombosis [71]. The contact activation (intrinsic) pathway of coagulation is initiated by negatively charged surfaces (DNA, etc.), which facilitate the activation of FXII. One of the results of FXIIa production is FXI cleavage to generate FXIa. FXIa initiates a series of Ca^2+^-dependent proteolytic events, leading to the generation of FIXa. FIXa and FVIIIa form tenase complexes (FIXa-FVIIIa), which convert FX to FXa. In this process, exposed PS on MPs provides binding sites for FVIII [72]. During the coagulation cascade, both the intrinsic and extrinsic coagulation pathways converge to the activation of FX. Subsequently, the formation of prothrombinase complexes (FXa-FVa) induces a burst of thrombin generation, triggering fibrin production and the activated platelet cascade and leading to plugging formation and bleeding cessation at the site of injury. PS on the outer leaflet contributes to the complex coagulation assembly and thrombin formation. Specifically, PS binds to the 9-12-carboxyglutamic acid domain at the NH2-terminus of coagulation factors (FII, FVII, FIX, FX) in the presence of Ca^2+^, and improves the catalytic efficiency of the prothrombinase complexes [73]. Consequently, PS on MPs plays a significant role in tumor-associated coagulation. 

Several control mechanisms are implemented in the coagulation network to avoid spatially inappropriate or excessive hemostasis, which could result in vascular occlusion, thrombus formation, and ischemia [74]. Thrombin can trigger protein C (PC)-activated anticoagulant circuits to limit local and temporary coagulation cascade. PC in circulation is activated by the binding of thrombin to thrombomodulin (TM) on the endothelial surface. Activated protein C (APC) binds to its cofactor (protein S) on the surface of activated platelets, inactivating FVa and FVIIIa. In large vessels, the endothelial protein C receptor localizes PC to the endothelial surface, where it can be activated by nearby thrombin-TM complexes [75]. Additionally, tissue factor pathway inhibitor (TFPI, a kunitz-type inhibitor) binds to the active site of FXa, and then interacts with the TF-FVIIa complexes, thereby suppressing their activity. 

Tumor cells can induce platelet activation in their vicinity in diverse ways, causing hypercoagulability and increased risk of thrombosis in cancer patients. Cancer cells release soluble mediators, including high mobility group box protein 1 (HMGB1), which connect with TLR4 and provoke local platelet activation [76]. Many tumors (i.e., pancreatic cancer) also secrete thrombin (a potent platelet activator), which mediates the interactions between platelets and coagulation pathways. Thrombin induces platelet activation and aggregation through three different receptors: PAR-1, PAR-4, and glycoprotein (GP) Ibα of GP Ib-IX-V complexes [77]. Following platelet activation, platelets and tumor cells utilize various adhesion receptors to form aggregates. Several interactions result in the tight binding of platelets to tumor cells, including (i) the binding of C-type lectin-like receptor 2 (CLEC-2) on platelets with podoplanin on tumor cells and (ii) the binding between p-selectin on platelets and Sialyl–Lewisx-conjugates on tumor cells, and (iii) integrin αIIbβ3 on platelets and αVβ3 on tumor cells, both of which bind to fibrin [72]. Cancer cells activate platelets and induce tumor cell-induced platelet aggregation (TCIPA) (Figure 2). Under the integrin α2β1, platelets adhere to collagen through GPVI, inducing the activation of SRC-family kinase and spleen tyrosine kinase (SYK), thereby resulting in stimulation of phospholipase Cγ and elevated Ca^2+^ levels in the cytoplasm through a series of downstream events involving small GTPase regulatory proteins [78]. Platelets experiencing prolonged Ca^2+^-dependent signaling will swell, undergo membrane ballooning, and expose PS on their membrane surface, due to activation of the Ca^2+^-activated scramblase (TMEM16F) mediated by the intracellular protease calpain 2 [79]. Platelets can induce thrombin generation by providing a membrane surface enriched for the anionic phospholipid PS. Integrin αIIbβ3 opens and binds to fibrinogen (acting as a bridge between platelets) and then activated FXIII cross-links and stabilizes fibrin, further forming aggregates and tumor cell-platelet embolism. Platelet-derived polyphosphate (poly-P) can activate the intrinsic coagulation pathway in an FXII-dependent manner. Platelets can also attract more circulating PLTs to the growing thrombus through granule secretion (both ADP and ATP) and thromboxane A2 release. 

## 4. PS Involvement in Tumor Metastasis by Mediating Coagulation

Infiltration of adjacent tissues and metastasis to distant organs are the primary features of malignant tumors. Distant metastases, rather than the primary tumor, are the leading cause of death in most cancer patients. Firstly, we elaborate on the general process of tumor metastasis, including enhanced metastatic properties by obtaining EMT, entry of tumor cells into the circulation through depredating the extracellular matrix by MMPs, immune escape through interacting with blood cells (platelets, etc.), adhesion to endothelial cells and subsequent extravasation by utilizing coagulation and establishing a vascular network. Before transfer, cancer cells obtain several phenotypic changes (e.g., mesenchymal phenotype), downregulate the adhesion of neighboring cells, promote changes in the cytoskeletal structures, and display metastatic properties such as enhanced mobility and invasiveness [80]. Subsequently, they leave the primary tumor as individual cells or collective groups (epithelial–mesenchymal transition (EMT) or mixed EMT) and localize along matrix proteins (guided to blood vessels or lymph through oxygen concentration detection). Cancer cells require the upregulation of some enzymes (MMPs) to degrade the extracellular matrix (ECM), communicate with cells within the matrix, and induce tumor-friendly switching of stromal cells (macrophages, myofibroblasts, etc.) [81]. The disseminated tumor cells in circulation are termed circulating tumor cells (CTCs), which protect themselves from immune attack (cytotoxic T-cells, natural killer (NK) cells) using the coagulation cascade and recruiting leukocytes and platelets. CTCs adhere to ECs by utilizing coagulation within the venules and express extravasated homing molecules. Exudative CTCs lose their migratory capabilities, and recover tight contact with other cells, ultimately boosting their growth [82]. In the metastatic site, cancer cells establish vascular networks to sustain an adequate supply of nutrients and oxygen and communicate with stromal cells, forming a good premetastatic niche. It is well known that metastasis is not a random process. The homing theory states that organs distant from the primary tumor site actively attract malignant cells through the expression of adhesion receptors and the secretion of soluble chemokines. Adhesion receptors or molecular addresses on ECs in vascular beds of distal organs that specifically capture CTCs support the active arrest view of the homing theory [83]. The “seed and soil” theory postulates that different organ environments provide different growth conditions for CTCs.

Normally, platelets and coagulation components are separated from tumor cells. However, over time tumor vessels lose their gatekeeper function and allow the extravasation of platelets and coagulation factors in the following cases. Firstly, the tumor volume approaches preexisting vessels owing to its volume-demanding growth [84]. VEGF, especially VEGF-A, as well as other chemokines from the TME, can induce increased permeability of blood vessels and activate ECs, which may lead to increased leukocyte extravasation [85]. Secondly, owing to excessive and persistent proangiogenic signals, tumor-associated blood vessels usually have abnormal morphologies characterized by excessive vessel branching, abnormal bulges and blind ends, discontinuous endothelial cell linings, as well as basement membrane and pericyte coverage defects [86]. Therefore, irregular angiogenesis becomes another avenue for blood components to leak into tumor masses. Vasculogenic mimicry (VM) is a third route for the extravasation of blood components into tumor tissue. This leakage allows intravascular platelets and coagulation factors (including thrombin, fibrinogen, etc.) to enter the extravascular space, where platelets are activated, and fibrinogen is converted to fibrin by coagulation activation, affecting tumor spread and metastasis in the tumor microenvironment.

Finally, coagulation components promote tumor growth and metastatic spread. The underlying mechanisms by which coagulation factors promote tumor growth, invasion, and metastasis have been a hot topic in cancer research. Most cancers, especially colon, ovarian, and lung, are generally recognized to have a biological interrelationship between cancer cells and the hemostatic system [87]. Malignant diseases correlate strongly with coagulation abnormalities and thrombus formation. This relationship is based upon the fact that cancer causes a prothrombotic switch of the body’s coagulation system; in turn, coagulation activation stimulates tumor growth and metastatic spread. The results of laboratory tests indicate that the process of fibrin formation and clearance is parallel with the development of malignant tumors and that fibrin and other coagulation products are essential for tumor progression [66]. Clinical research suggests that the expression of pro-coagulants on cancer cells is a prognostic indicator of metastasis and poor outcome. The emerging literature supports that hemostatic components are crucial factors affecting the prognosis of breast cancer [8]. The critical steps of tumor progression, including tumor cell survival and proliferation, are partly mediated by the components of the hemostatic system [88,89]. The most reported coagulation proteins involved in cancer progression include TF, TF-FVII, thrombin, and fibrin. Given the intimate connections between cancer and coagulation, PS-mediated coagulation is also involved in tumor progression. We will further discuss the interplay between tumor progression and PS-dependent coagulation by focusing on angiogenesis, EMT, ECM remodeling, tumor extravasation, and immune escape.

### 4.1. PS-Mediated Coagulation Cascade Involvement in EMT

The epithelial–mesenchymal transition (EMT) is the activation of a portion of primary tumor cells, resulting in the cancer cells losing their characteristic apical–basal polarity and acquiring some properties of mesenchymal cells. Cancer cells undergoing EMT display morphological and molecular alterations, such as up-regulation of mesenchymal markers (N-cadherin, etc.) and down-regulation of epithelial markers (E-cadherin) [80]. Additionally, tumor cells acquire a hybrid epithelial (adhesive)/mesenchymal (migratory) phenotype, which protects CTCs clusters in the blood from being destroyed when transitioning between epithelial and mesenchymal phenotypes [90]. Evidence suggests that compared to individually migrating CTCs, these clusters have more potent anti-apoptotic activity, more efficiency exiting the bloodstream, and greater tumor-initiating potential [91,92]. The key events of EMT are the dissolution of epithelial cell junctions, restoration of front-rear polarity, alteration of cytoskeletal structures, and increased cell motility. Multiple signaling pathways participate in EMT regulation, and these include transforming growth factor (TGF), Wnt/β-catenin, and Notch pathways [93]. In addition to promoting the transformation of primary tumors into aggressive malignant tumors, EMT also involves the generation of cancer stem cell-like characteristics, such as enhanced self-renewal capacity, tumor-initiating ability, resistance to apoptosis, and chemoresistance. 

EMT and TGF-β signaling genes are enriched in gastric cancers with high coagulation scores, suggesting an association between coagulation and expression levels of the EMT gene. The view of TF involved in EMT is being increasingly accepted [94]. The downstream events of TF activation consist of thrombin generation, platelet activation, and facilitating metastasis by EMT. TF regulates cytoskeletal remodeling and enhances tumor cell migration. The TF-blocking antibody (CNTO 859) delays A431 cell initiation and metastasis by blocking EMT [95]. Thrombin can induce EMT in various cancers. Thrombin-mediated PAR-1 activation induces the reprogramming of gene expression by stimulating transcription factors like SNAIL1, which has been shown to drive EMT in embryogenesis [96]. In breast cancer, thrombin/ PAR1 complexes cause changes in components of basement membranes (increased expression of β-catenin and Wnt, and decreased expression of E-cadherin) as well as the alteration in cytoskeletal proteins (myosin IIA), which together regulate the EMT engaged in tumor progression [97,98]. During pulmonary metastasis, platelet-derived TGF-β synergizes with thrombin-activated platelet/tumor cell complexes and stimulates TGF-β/Smad and nuclear factor-κB (NF-κB) signaling pathways, leading to the transition to invasive mesenchymal-like phenotype and enhanced metastasis in vivo [99]. In patients with NSCLC, thrombin induces the formation of VM (tumor cells transform into ECs via EMT) by PAR1-mediated NF-κB signaling cascade and accelerates tumor progression, and increases resistance to treatment [100]. 

#### HIF Involvement in EMT

PS-mediated coagulation activation and thrombosis in the tumor microenvironment (TME) leads to localized ischemia and hypoxia and necrosis of tumor tissue. Hypoxia is a significant microenvironmental factor during metastatic spreading. Compared with severe hypoxia, mild to moderate hypoxia has greater effects on tumor progression since tumor cells are more vulnerable to death in severe hypoxia [101]. Hypoxia signals activate the pre-migratory and invasive phenotype of tumor cells via multiple mechanisms. Hypoxia-inducible factor-1α (HIF-1α) is a key transcriptional regulator of hypoxia. HIF1α promotes EMT through the enhancement of signaling pathways, including Snail, Twist, Notch, and β-catenin, thereby resulting in cancer cell invasion, migration, and tumor metastasis. Curcumin suppresses colon cancer growth and metastasis, associated with upregulated epithelial markers and downregulated mesenchymal markers as well as EMT-related transcription factors Snail and Twist [102]. Chen et al. showed that the anticancer effects of melatonin on osteosarcoma OA MG-63 cells were mediated by the inhibition of EMT through the HIF-1α/Snail/MMP 9 signaling pathways [103]. Prostate cancer stem cells with a mesenchymal phenotype are triggered by cancer-associated fibroblasts (CAFs) by HIF-1α/β-catenin-dependent signaling pathways [104]. The expression and secretion of midkine (MDK) protein in lung cancer cells is regulated by HIF-1α. More specifically, MDK mRNA expression is upregulated under hypoxic conditions. MDK protein secreted by NSCLC cells interacts with Notch 2, activates Notch signaling pathways, induces EMT, upregulates NF-κB, and enhances pro-tumor effects [105]. HIF-1α can also mediate tumor cells’ adaptation to the hypoxic microenvironment, accelerate EMT, and increase resistance to radiotherapy/chemotherapy. Overexpression of pre-mRNA processing factor 4B (PRP4B) activates miR-210 transcription in a HIF-1α-dependent manner, thereby activating p53 and contributing to EMT and drug-resistant induction [106]. The p65 subunit of NF-κB or HIF-1α is downregulated by siRNA to reverse the EMT phenotype under hypoxic conditions in vitro, inhibits the proliferation of pancreatic cancer cell lines (PANC-1, BxPC3), induces apoptosis, and enhances the efficacy of gemcitabine in treating pancreatic cancer [107]. The activation of HIF-1α and NF-κB loops under hypoxic conditions is mechanistically implicated in chemotherapeutic drug-resistant phenotype (EMT phenotype). HIF-2α is also involved in EMT and chemotherapy resistance. In hypoxic cultured CFPAC-1 and BxPC-3 cells, HIF-2α increases the expression of miR-301a, and miR-301a overexpression can directly target downstream TP63 and promote hypoxia-induced EMT in pancreatic ductal adenocarcinoma (PDAC) cells [108]. HIF2α can also regulate EMT through activating Wnt and Notch pathways and facilitate stem cell phenotype and chemoresistance in breast cancer cells [109]. 

### 4.2. PS-Mediated Coagulation Cascade Involved in ECM Remodeling

Extracellular matrix (ECM) remodeling, including protease degradation of the matrix, is critical for tumor cells invading the surrounding stroma and the extravasation of blood vessels. Typically, ECM is composed of multiple proteins, such as diverse types of collagens (I, III, IV, V), fibronectins, and laminins. The MMP family degrades ECM and catalyzes proteolysis of the basement membrane, thereby enhancing the mobility of migrating tumor cells and making them accessible to the circulatory system. Recently, studies on the activities of MMPs and their tissue inhibitors (TIMPS) in aggressive tumors have shown that the balance between MMPs and TIMPs contributes to maintaining ECM homeostasis. Conversely, the imbalance promotes cancer metastasis by modifying the ECM environment. The combination of TF with FVIIa stimulates the activation of several intercellular signaling pathways (mitogen-activated protein kinase (MAPK), phosphatidylinositol 3-kinase (PI3K), AKT), ECM remodeling, and cell proliferation [110,111]. In breast cancer cells, TF/FVIIa trypsin-mediated PAR2 activation facilitates MMP2 expression by PI3K/AKT-NF-ĸB signaling pathways, followed by NF-κB inhibitory protein α degradation and NF-ĸB translocation into the nucleus [112]. Secreted MMP-2 cleaves ECM to facilitate tumor invasion. Thrombin can promote trans-basement membrane migration through ECM remodeling following activation of type IV collagen degrading enzymes, MMP-2 and αvβ3 integrin [113]. In nasopharyngeal carcinoma, thrombin-induced PAR-1 activation increases the expression of MMP-2 and MMP-9, thus facilitating ECM degradation and destruction of basement membrane by tumor cells [114]. Additionally, thrombin can activate platelets to support matrix degradation. Activated platelets promote ECM remodeling by releasing MMPs from α-granules [115]. 

#### Plasminogen Activator/Plasmin System

Apart from the MMP family, the plasminogen activator/plasmin system is also associated with tumor invasion and metastasis. Plasmin initiates the response of ECM remodeling, including two plasminogen activators, urokinase-type plasminogen activator (uPA) and tissue-type plasminogen activator (tPA). The fibrinolytic activity of uPA performs a predominant role in cell migration and tumor metastasis and is tightly regulated by proteolytic cleavage. The binding of uPA to its specific receptor uPAR converts plasminogen into plasmin, which degrades fibrin and other ECM components, including type IV collagen, fibronectin, and laminin [116]. Moreover, the uPA-catalyzed plasmin generation promotes the conversion of MMP zymogens to their active form and degrades matrix components. Proteolysis of ECM enables the release of tumor cells from their adhesion sites, thus permitting cell migration, manifesting as invasion at local sites and metastasis at distant sites. Multiple studies have demonstrated the expression of active uPA in tumor cells is related to their invasive potential [83,117]. In pancreatic cancer patients, post-operative survival is shorter in patients with simultaneous overexpression of uPA and uPAR compared to those with only uPA or its receptor overexpression [118]. Coagulation factors can modulate the expression u-PA/u-PAR to enhance tumor aggressiveness. TF pathways can upregulate the expression of uPAR, which can promote the production of matrix-degrading enzymes. It has been reported that the formation of TF/VIIa can upregulate u-PAR expression in the human pancreatic cancer cell line, thereby enhancing tumor invasion and metastasis via u-PA/u-PAR pathways [119]. Furthermore, the addition of human α-thrombin to PC-3 prostate cancer cell cultures produces dose-dependent and time-dependent increases in u-PA secretion by functional thrombin receptors on PC-3 cells [120].

### 4.3. PS-Mediated Coagulation Cascade Involved in Immune Escape

In circulation, CTCs can encounter potential barriers, including anoikis (programmed cell death owing to loss of cell adhesion) and mechanical shear forces that physically hinder tumor cell anchoring. Tumor cells can also be recognized and eliminated by immune cells (NK cells, etc.). Furthermore, in contrast to the immunosuppressed primary tumor microenvironment, immune cells can more effectively attack and destroy tumor cells in circulation. The interactions between the tumor and immune cells eventually determine whether cancer can be contained locally or escape and invade distant sites. However, circulating CTCs can adopt mechanisms of immune evasion to silence their immunogenicity and activate immunosuppressive pathways to evade anti-tumor efforts. Currently, multiple mechanisms, including antigen deficiency, lack of costimulatory signals on the tumor cell surface, and tumor cell-induced immunosuppression, have been proposed [121,122]. CTCs regulate autoantigenicity by antigen endocytosis or shedding [123]. Generally, presented tumor antigens in the absence of costimulatory signals cause T cells to generate antigen tolerance. Programmed death-1 (PD-1) and its ligand are important checkpoint proteins to regulate antitumor immune responses. CTCs expressing PD-L1 are shown to limit T cell function and promote immune tolerance by binding to PD-1 on immune cells [124,125]. CTCs produce immune inhibitory molecules (indoleamine 2,3-dioxygenase, etc.), which directly restrain immune responses or recruit Tregs that secrete immunosuppressive cytokines [126]. Small parts of CTCs interact tightly with macrophages, CAFs, or myeloid-derived suppressor cells (MDSCs) to escape from the immune system and promote their survival [127]. Furthermore, by forming NETs, circulating neutrophils can inhibit peripheral leukocyte activation, NK cell function, the antitumor response of effector T cells, and even cooperate with other immune cells (e.g., IL17-producing γδ T cells) to assist the evasion of CTCs from immune surveillance [128,129].

TF is composed of full-length TF (flTF, FIII) and, alternatively, spliced TF (asTF). Once cancer cells enter the blood, flTF-mediated activation of coagulation pathways exerts an essential role in immune escape. flTF prevents CTCs from being cleared by NK cells via thrombin-dependent mechanisms and further supports tumor development and metastasis. Additionally, one of the key mechanisms by which the TF-thrombin-PAR-1 signaling axis in tumor cells facilitates PDAC growth and disease progression is the suppression of antitumor immunity [130]. Further studies of immune cell depletion show that in C57Bl/6 mice, CD8 T cells but not CD4 T cells or NK cells mediate the elimination of KPC-PAR-1 tumor cells [130]. Once generated, thrombin can directly cleave complement component C5, subsequently generate C5a and C5b, and reach the TME by the leaky vasculature. C5a, namely anaphylatoxin, results in an immunosuppressive microenvironment by the recruitment of MDSCs, which provides new insights on coagulation-induced complement activation promoting tumor progression [131] (Figure 2). Additionally, FXIII-stabilized platelet/fibrin thrombi impede the approach of NK cells toward tumor cells by forming a physical barrier. Experimental and spontaneous metastasis is reduced when the fibrinogen or fibrin cross-linked FXIII genes are knocked out [132]. Similarly, the deletion of fibrinogen or fibrin cross-linking FXIII can strongly reduce metastasis in an NK cell-dependent manner [89]. 

Platelets are not just bystander cells in circulation but are functional players in many steps of primary tumor growth and metastasis. The interactions of tumor cells with platelets and the process of tumor cell-platelet embolism formation have been described above. Tumor cell-platelet embolism protects CTCs from high fluid shear force and hosts immune surveillance. Interestingly, platelets appear to induce the resistance of CTCs to anoikis, which is mediated by the activation of the yes-associated protein-1 (YAP1) signaling pathway [133]. Platelets can also interfere with the recognition of tumor cells by NK cells. Platelets transfer normal MHC class I-like molecules onto the tumor cell surface, which makes tumor cells unrecognized as foreign cells [134]. TNF family member (the glucocorticoid-induced tumor necrosis factor receptor ligand, GITRL) released from α-granules of activated platelets is co-expressed with p-selectin and induces significantly reduced NK cell-mediated cytotoxicity and IFN-γ release [135]. Platelet-derived TGF-β can downregulate NKG2D and inhibit the anti-tumor reactivity of NK cells, thereby reducing their activity [136]. Additionally, platelet-derived VEGF can also impair the maturation of dendritic cells and their interactions with B and T lymphocytes and NK cells [137] (Figure 2). 

### 4.4. PS-Mediated Coagulation Cascade Involved in Extravasation of CTCs

Recent studies have revealed that the interactions of CTCs with the hostile blood microenvironment are critical for endothelial cell adhesion and tumor metastasis. Extravasation constitutes a multistep phenomenon that includes endothelial cell adhesion at secondary sites, regulation of the endothelial barrier, as well as trans-endothelial migration into the underlying tissue. Rolling and binding of leukocytes and/or tumor cells, mediated by the selectin family of adhesion molecules, initializes the extravasation process. Following selectin-mediated rolling, integrins are activated, bind to their endothelial ligands, and allow for tight adhesion of leukocytes. Tumor cells adhere to the vascular endothelium utilizing mechanisms used by leukocytes. Integrin αLß2 (lymphocyte function-associated antigen LFA-1) is expressed on lymphocytes and neutrophils, whereas integrin αMβ2 (Mac-1) is predominantly expressed on neutrophils [138]. Integrin αLß2 has been shown to be a ligand for intercellular adhesion molecule 1/2 (ICAM-1/2) and junctional adhesion molecule A (JAM-A). Integrin αMβ2 can interact with ICAM-1/2 and JAM-C on ECs. Tumor cells do not express the above-mentioned integrins, but they can express ICAM-1 and adhere to the endothelium using leukocytes as linker cells. Tumor cells interact with leukocytes via ICAM-1/LFA-1 (αLß2), which in turn bind to ICAM-1 on the ECs via LFA-1 [139]. Other integrin-mediated tumor cell adhesion includes αVβ1 or αVβ3 on tumor cells with L1 cell adhesion molecular (L1CAM) on the endothelium and α4β1 on tumor cells with vascular cell adhesion molecule-1 (VCAM1) on the endothelium, further allowing CTCs to extravasate and form distant metastases (Figure 2). There are two pathways for trans-endothelial migration of malignant cells: paracellular (i.e., via cell–cell junctions) and transcellular (i.e., via individual ECs). Multiple immune cells mediate the extravasation of CTCs via different mechanisms. Neutrophils are attracted to sites of metastasis through the CXCR2 receptor on their surface, binding to chemokine (C-X-C motif) ligand 5 (CXCL5) and CXCL7 released from activated platelets [140]. The presence of leukocytes in tumor cell-platelet microthrombi significantly promotes the extravasation of tumor cells and supports tumor spread and metastasis. NETs can isolate CTCs mediated by β1 integrins and promote adhesion to the endothelium before or during extravasation [141]. Tumor cells directly or indirectly recruit inflammatory monocytes by releasing chemokines (cc-chemokine ligand 2 (CCL2)) or inducing local endothelial activation (E-selectin expression) [142,143]. CCL2 and E-selectin themselves can also directly facilitate the extravasation and metastasis of tumor cells. Once recruited to the metastatic site, monocytes differentiate into metastasis-associated macrophages (MAMs) within the underlying tissue and then induce tumor cell extravasation by VEGF-mediated increase in vascular permeability [144].

CTCs must be near endothelial junctions to exert their effects, and shear forces in the blood make it impossible for tumor cells to be in prolonged contact with the endothelium; thus, endogenous thrombin generation is essential for the rapid response of ECs. Thrombin-mediated integrin activation and ICAM-1 expression promote the interactions of CTCs with leukocytes, platelets, and ECs, improve the survival of CTCs in the bloodstream, and promote their microvascular retention under flow conditions [79,145]. Due to the coagulation activity of thrombin, fibrinogen is converted to fibrin, supporting adhesion events between tumor cells and other molecules. Studies show that thrombin-dependent fibrin(ogen) deposition is an important determinant of cancer metastasis and promotes stable adhesion and survival of metastatic emboli following cancer-cell implantation. Fibrinogen (Fg) and soluble fibrin (sFn), ligands for integrin αvβ3 and ICAM-1, respectively, are expressed in melanoma and epithelial tumor cells. Under shear flow, sFn and Fg enhance polymorphonuclear neutrophil (PMN)-mediated melanoma adhesion to the endothelium via ICAM-1 [146]. They also bind to CD11b/CD18 (Mac-1) on PMN, thereby enhancing melanoma-PMN aggregation and adhesion to ECs [147]. 

Studies indicate an association between thrombin generation and trans-endothelial migration of malignant cells induced by disruption of the endothelial barrier. The endothelial gap of blood vessels is composed of two interrelated processes, namely, the loss of integrity of VE-cadherin (VEC) complexes and the obstruction of the endothelial membrane by the actomyosin machinery. Thrombin increases endothelial Ca^2+^ concentration by PAR-1 activation and activates the C-sensitive PKC pathway and subsequent tyrosine phosphatase, leading to the disassembly of local adhesion junctions (VEC) and increased permeability of the endothelial barrier [148]. Additionally, thrombin-mediated p21-activated kinase (PAK) activity also increases endothelial permeability, which is related to myosin phosphorylation and the appearance of paracellular pores [149] (Figure 2). Another mechanism of thrombin-induced disruption of the endothelial barrier is PAR1 /PKC/Ca^2+^-dependent activation of actin stress fibers. Under thrombin stimulation, the formation of actin stress fibers and activation of the actomyosin contractile machinery lead to circular morphology of ECs and subsequent loss of cell–cell adhesion, which in turn promotes trans-endothelial migration and tumor extravasation [150,151]. 

Platelets contribute to the adhesion and extravasation of tumor cells. Adhesion of CTCs to platelets is mediated by platelet adhesion receptors, including integrins p-selectin and αIIbβ3, thereby supporting firm adhesion of CTCs to the endothelium [152]. Platelets also increase the permeability of blood vessels to promote tumor cell extravasation. Platelets activated by tumor cells release ATP from dense granules, induce the activation of endothelial P2Y2 receptors and allow trans-endothelial migration of tumor cells by increasing the permeability of blood vessels [153] (Figure 2). Subsequently, the disrupted endothelial barrier facilitates tumor cell translocation and extravasation and the formation of metastatic foci. Mammadova-Bach E et al. revealed that the interactions between integrin α6β1 and its receptor on platelets and MMP9 on CTCs are required for the extravasation process of cancer cells [154]. Additionally, α6β1 on platelets binds to ADAM9 on tumor cells, facilitating platelet activation and tumor cell extravasation [155]. 

### 4.5. PS-Mediated Coagulation Cascade Involved in Angiogenesis

Angiogenesis is essential for tumor growth and metastasis. The formation of new blood vessels can occur by the sprouting of new blood capillaries (angiogenesis) or division into multiple daughter vessels (intussusception) from the existing vessels. Angiogenesis consists of a cascade of sequential processes emanating from microvascular ECs that are stimulated to proliferate and degrade the endothelial basement membrane, migrate, penetrate host stroma, and form capillary sprouts [83]. The recruitment of ECs during angiogenesis depends on the breakdown of basement membranes, which occurs under the control of overexpressed activators, including VEGF and angiopoietin. [156]. These VEGF signaling pathways control the angiogenic processes, and their dysregulation is related to metastatic colorectal cancer and NSCLC. Apart from the enhancement of clot formation, coagulation factors also have pro-tumorigenic potential, including promoting angiogenesis. Coagulation factors involved in tumor angiogenesis include TF, FVII, thrombin, thrombin receptor (protease-activated receptor), fibrin, and FXIII. 

#### 4.5.1. The Role of TF in Tumor Angiogenesis

TF engages in cancer development, specifically tumor cell proliferation, survival, and metastasis. In certain malignant tumors, elevated TF expression is observed in both serum and tumor tissues [157,158]. Interestingly, circulating TF-positive MPs or MP-TF activity in tumor tissue correlates with microvessel density. TF promotes tumor-directed angiogenesis by upregulating vascular VEGF expression and downregulating the angiogenesis inhibitor (thrombospondin) expression. The cytoplasmic structural domain of flTF is important for controlling VEGF expression, and its phosphorylation is related to VEGF expression in tumors. In the coagulation-dependent pathways, the interactions of TF with FVIIa cause Ca^2+^ oscillations and altered gene expression, while the formation of TF/VIIa complexes induces activation of the MAPK pathway, which is a major pathway of VEGF expression [159]. Additionally, flTF/FVIIa-mediated PAR2 activation also results in the generation of pro-angiogenic factors, including VEGF, IL-8, and CXCL1 [160]. TF/VIIa facilitates PAR-1- and PAR-2-mediated signaling responsible for the proliferative response of cancer cells. The effect of PAR-2-mediated angiogenesis is superior to that induced by PAR-1 signaling. PAR2 signaling has been shown to act in the initial stages of tumor angiogenesis, the so-called angiogenic switch. The catalytic activity of TF-FVIIa complexes contributes to FX activation, and the TF/FVIIa/FXa complexes activate one or more PARs, such as endothelial PAR-2, to support angiogenesis in vivo [161] (Figure 3). Nude mice carrying PDAC cells that overexpress asTF produce larger tumors and increased angiogenesis than those carrying overexpressing flTF cells [162]. Elevated asTF facilitates tube formation in A549 cells by increasing VEGF, Cyr61, and CCL2. Unlike the flTF-PAR interaction, asTF elicits adhesion kinase (FAK), p38 MAPK, and Akt phosphorylation by interacting with integrins β1 and β3 on ECs, having potent pro-angiogenic properties [163]. 6B4, an antibody that interferes with TF-integrin interactions, effectively inhibits the pro-angiogenic properties of asTF.

#### 4.5.2. The Role of Thrombin in Tumor Angiogenesis 

Notably, several studies have also shown that thrombin can promote angiogenesis. Firstly, the direct interaction between thrombin and tumor cells triggers the development of new blood vessels. Secondly, thrombin can also inhibit apoptosis and induce the proliferation and differentiation of vascular progenitor cells. Thirdly, thrombin-induced angiogenesis is related to the overexpression of multiple proangiogenic factors, mainly VEGF. Thrombin can increase VEGF secretion in DU145 prostate cells, human FS4 fibroblasts and CHRF megakaryocytes [164]. In gliomas, thrombin improves the levels of VEGF by the activation of HIF-1α and p44/42 MAPK pathways [165]. Thrombin can upregulate the expression of ANG-2, VEGF, and their receptors (KDR and Flt1) and significantly enhance VEGF-induced EC proliferation [166]. Cancer cells secrete VEGFA, which initiates tumor angiogenesis through binding to VEGF receptor 2 (VEGFR2) expressed on adjacent vascular ECs. The gradient of soluble VEGFA can cause the formation of motile ECs, namely tip cells, which break down the surrounding ECM and result in the growth of new vascular sprouts toward VEGFA [86]. Furthermore, VEGF-induced vascular hyperpermeability enhances the extravasation of plasma proteins and neo-vascularization [167]. 

Thrombin serves pro-angiogenic roles via PAR-dependent mechanisms. Thrombin/ PAR-1 activation affects tumor angiogenesis in medulloblastoma and high-grade brain tumors. Thrombin/PAR-1 activation produces strong mitogen activity on ECs and vascular progenitor cells to induce EC tubular formation in the matrix membrane. Furthermore, Thrombin/PAR-1 activated platelets lead to their aggregation and degranulation (releasing vascular growth factors, including VEGF and PDGF, Figure 3). VEGF is synthesized by megakaryocytes and released from mature platelets, following activated by thrombin, promoting the thickening of tumor vasculature. In patients treated with anticoagulation (heparin), platelets show reduced VEGF release and angiogenic potential, suggesting that anticoagulant therapy attenuates the angiogenic potential of platelets [168]. Additionally, following thrombin-mediated platelet activation, PAR-1 signaling pathways result in the synthesis and release of thromboxane 2 and 12(S)-hydroxyeicosatetraenoic acid. These substances promote angiogenesis by acting on specific receptors (TPα, 12HETER1/GPR31) and transcriptional regulation of factors, including HIF1α and VEGF [169,170]. 

Thrombin-induced inflammatory response upregulates the expression of cytokines, angiogenic factors, and MMPs, promoting angiogenesis and metastasis. Osteopontin (OPN) cleaved by thrombin induces increased proteolytic activity, tumor growth, colony formation, and angiogenesis in cancer cells compared to untreated controls [171]. Furthermore, thrombin increases the expression and secretion of growth-related oncogene α (GROα), which is expressed in human MCF-7 breast cancer cells and PC3 prostate cancer cells. Thrombin and GROα upregulate the expression of MMP and angiogenic molecules, which are critical for angiogenesis and tumor progression [172]. Hu et al. showed that thrombin supported angiogenesis by upregulating the aspartic lysosomal protease (cathepsin D, CD) in six different tumor cell lines and human ECs. The CD is recognized to be a proangiogenic molecule and induces angiogenesis by proteolytic activation of MMP-9 [173]. 

#### 4.5.3. The Role of Fibrin and FXIII in Tumor Angiogenesis

The combination of ECs with fibrin and the involvement of adhesion molecules (vascular endothelial cadherin) are required for capillary formation, which is a crucial step in angiogenesis. Tumor-containing tissues show that VEGF-induced vascular leakage stimulates increased fibrin deposition to promote angiogenesis [174]. The fibrin matrix around the tumor cells provides a temporary pro-angiogenic environment to support the formation of blood vessels and stimulate EC proliferation and migration. The fibrin matrix also upregulates the expression of αVβ3 integrin receptor to promote angiogenic responses. Additionally, the fibrin matrix binds and sequesters numerous growth factors (VEGF, etc.) and protects platelet-derived VEGF and thrombin from being degraded by proteinase in ECM [175]. Fg and fibrin fragments, such as the E fragment, are shown to stimulate angiogenesis both in vitro and in vivo. In experiments, FXIII is identified as a potential regulator of angiogenesis in vivo. FXIII supports angiogenesis by interacting with substrates (fibrin, fibronectin) with pro-angiogenic properties [132]. Finally, FXIIIa may indirectly control angiogenesis by supporting the stabilization of platelet thrombus and local release of pro-angiogenic factors from activated platelets. 

## 5. Treatment Strategy

### 5.1. Blocking PS on Tumor Cells and Their MPs

An activated coagulation cascade contributes to tumor spread and metastasis, so inhibition of the PS-mediated coagulation pathways decelerates tumor progression. A common drug targeting PS is Annexin V, which binds specifically and stably to external receptors or binding sites in tumor vascular ECs. Systemic administration of Annexin V or other PS ligands can inhibit tumor progression by blocking the tumor-supporting properties of apoptotic cells and tumor-derived MPs [57]. Interestingly, the binding of lactadherin to PS is more sensitive than that of annexin V to PS [176]. Lactadherin facilitates binding to PS in a stereotactic and calcium-independent manner via vesicle bending. Additionally, lactadherin effectively competes with PS for coagulation binding sites (FV and FVIII) to exert anticoagulant effects. Multiple studies have demonstrated that lactadherin is significantly better than Annexin V in coagulation inhibition. The development of a PS-targeting antibody (bavituximab) is currently being evaluated in multiple clinical trials, and clinical studies are planned to assess the efficacy of bavituximab with anti-PD-L1 antibodies for the treatment [177,178]. 

### 5.2. Cancer Treatment Strategies Targeting Thrombosis

Activated platelets release MPs and externalize PS, and thus inhibition of platelet activation can reduce the PS externalization-induced coagulation activation. Platelets play a predominant role in promoting CTC survival and tumor cell extravasation, suggesting that antiplatelet agents could be expected to prevent or reduce metastatic spread, as has been shown by various preclinical studies. Aspirin can reduce the incidence of colorectal cancer and cancer-related mortality. Thus, the U.S. Preventive Services Task Force recommends that healthy individuals take low-dose aspirin daily to prevent colorectal cancer. Studies report the association between aspirin use and the reduction of distant recurrence and improved survival with a minimum of one year [179]. Clinical trials, including the aspirin trial (NCT02804815), are ongoing to assess whether regular aspirin administration following early cancer treatment prevents recurrence and improves patient survival. It would be interesting if CTCs monitoring could be included in trials, as the use of aspirin may potentially reduce the platelet-rich thrombi associated with the initial stage of metastasis and prevent the occurrence of metastatic disease. In addition to aspirin, oral antiplatelet agents include ADP receptor antagonists (clopidogrel, prasugrel) and dipyridamole, which block adenosine uptake into platelets and inhibit cyclic AMP phosphodiesterase. These drugs inhibit platelet activation, are better tolerated, and have known safety profiles. The use of these drugs may be particularly attractive for improving survival in women with metastatic disease [180]. Platelet function can be suppressed by integrin αIIb inhibitor, which is reported to reduce the progression of metastatic cancer [181]. Highly specific GP αIIb β3 receptor antagonists inhibit platelet aggregation in vivo, thus affecting CTC function and downstream metastasis [182]. Since the contact between tumor cells and platelets can be mediated by CLEC-2-podoplanin interactions, targeting this early step could impede tumor cell-induced platelet aggregation cascade, thereby inhibiting the extravasation and metastases of CTCs [183]. Thus, CLEC-2 becomes an attractive target for delaying tumor progression [184].

In both in vitro and in vivo studies, anti-cancer therapy targeting coagulation factors has a profound impact on tumor behavior, including proliferation and metastasis. Consequently, targeting activated coagulation factors provides a feasible strategy for treating cancers. The coagulation properties of flTF aggravate cancer progression by enhancing metastatic potential. Currently, two potential flTF targeting agents are under clinical investigation: Altor Bioscience (a TF inhibitory antibody) and Pharmacyclics (a small-molecule inhibitor of FVIIa) [160]. TFPI-like inhibitors, NAPc2 and Ixolaris, have anticancer activities that are recapitulated by direct inhibitors of FVIIa. A TF-inhibitory antibody or FVII deficiency has been shown to reduce metastasis in a mouse breast cancer model [185]. Targeting asTF may suppress tumor spread by reducing angiogenesis, making asTF a new cancer target.

Atopaxar (PAR1 inhibitor) is an attractive therapeutic target and exerts effects via reduced platelet activation. Further preclinical and clinical studies are required to determine the efficacy of PAR1-targeted therapy in inhibiting PDAC progression. Since PAR2 signaling pathways are dependent on the formation of flTF/FVIIa or flTF/FVIIa/FXa complexes, reducing the effect of FVII and FX in cancer patients may provide insights into whether this indirect therapy of anti-flTF signaling pathways has the therapeutic potential [160]. A monoclonal antibody that suppresses TF signaling via PAR2 reduces tumor growth, while another TF-directed antibody that suppresses its proteolytic function only blocks bloodstream metastasis [186]. 

Thrombin is a main actor in cancer spread and can modulate the behavior of tumor cells directly by its receptors (PAR) and indirectly by the generation of a fibrin matrix. It is now recognized that thrombin promotes the interactions between tumor cells, platelets, and ECs, thereby seeding and metastasizing malignant cells. It thus has become an important therapeutic target for cancer patients. The treatment of dabigatran alone can prevent tumor-induced circulating MP increase and can reduce the number of tumor-induced activated platelets by 40%, indicating thrombin exerts essential effects in intravascular events [187]. The combination of dabigatran with gemcitabine inhibits primary tumor growth and prevents tumor cell spread from pancreatic cancer in mice [188]. Argatroban (a specific thrombin inhibitor) can also reduce the migration and metastatic bone potential of B16BL6 melanoma [189]. Additionally, Hirudin is a highly specific thrombin inhibitor that has been shown to significantly inhibit tumor entry into the bloodstream and spontaneous metastasis in a mouse model, resulting in prolonged survival time [190]. In a lung cancer model, r-hirudin and thrombin inhibitor peptides inhibit tumor progression, spread, and spontaneous metastasis by reducing VM formation [100].

As experimental studies increasingly reveal the anti-malignant properties of antithrombotic agents, evidence shows that adjuvant therapy with anticoagulants may improve the prognosis of cancer patients. Direct oral anticoagulant drugs (DOACs, dabigatran, apixaban, and rivaroxaban) alter treatment options. DOACs may also provide a potential therapeutic avenue for targeting CTCs [191]. These drugs can rapidly inhibit activated FX or thrombin. However, the use of DOACs in cancer patients is to be implemented with caution, mainly because both edoxaban and rivaroxaban have the risk for serious hemorrhage, especially in patients with gastrointestinal cancers. Because absorption of all DOACs occurs in the stomach or proximal small intestine, low-molecular-weight heparin (LMWH) is preferred for patients with drug interaction problems and for patients who have undergone upper gastrointestinal surgery. LMWH will continue to be used in patients with thrombocytopenia, recurrent VTE, central nervous system cancer, or severe renal impairment [192]. 

Heparin can be administered as LMWH (dalteparin, tinzaparin, and enoxaparin) or as unfractionated heparin. Owing to anticoagulant properties, heparin can inhibit cancer cell-induced thrombin and fibrin formation and intravascular blockage of cancer cells, thereby affecting metastasis [193]. Heparin may also influence cancer progression by modulating fltf-mediated signaling, but it is unclear to what extent flTF-specific signaling contributes to the role of heparin in cancer treatment. LMWH can prolong the survival of cancer patients. In six recently published randomized controlled trials, four diverse types of LMWH improved survival in patients with advanced cancer [194,195]. Patients treated with LMWH show a modest but significant survival benefit compared to those without LMWH [196]. 

### 5.3. Other Treatments

As mentioned above, cancer cells express some integrins that mediate adhesion to the vascular endothelium, facilitating extravasation and metastasis. Thus, pharmacological inhibition of these integrins may be a way to slow or even stop cancer progression. The drugs targeting integrins include antibody-based drugs (the most abundant type), small-molecule drugs, and peptide drugs [197]. Intetumumab (CNTO 95) is a human anti-αV integrin antibody that suppresses melanoma cell migration and invasion in vitro [198]. Abituzimab (DI17E6) is a humanized monoclonal antibody against αV subunit-containing integrins that suppresses the migration and invasion of prostate cancer cells [199]. The orally active αVß3 integrin inhibitor MK-0429 can reduce lung metastasis and melanoma burden in mouse models. MK-0429 is also utilized in patients with hormone-refractory prostate cancer and bone metastases [200]. D-pinitol modulates FAK, c-Src, and NF-κB pathways to inhibit αVß3 integrin and has anti-metastatic effects in human prostate cancer cells [201].

#### 5.3.1. Targeting HIF Therapy

HIFs mediate EMT via multiple signaling pathways, thereby resulting in tumor metastasis and chemoresistance. HIFs are promising therapeutic targets for the management of cancer. Cetuximab reduces HIF-1α mainly at the level of protein synthesis. RX-0047 (a potent inhibitor of HIF-1α) is a 20-mer phosphorothioate antisense oligonucleotide that directly suppresses HIF-1α by reducing the levels of mRNA and protein [202]. Non-specific inhibitors of HIF-2α usually inhibit HIF-2α transcription, protein synthesis, and nuclear translocation [203]. Among the small molecules currently being developed, TC-S 7009, PT-2977, and HIF-2α transcriptional inhibitors are reported as HIF-2α-specific antagonists [204,205].

#### 5.3.2. Targeting VEGF Therapy

Thrombin promotes tumor angiogenesis mainly through VEGF, which makes it an attractive target for anti-angiogenic therapy [206]. VEGF inhibitors block the activation of VEGFR receptors [207] and further intracellular signaling, thereby inhibiting endothelial cell proliferation and migration and preventing the formation of tumor vascular networks. Following FDA approval, bevacizumab is widely used in clinical practice in the United States. In glioblastoma diagnosis, one-third of patients have no tumor progression within 6 months following treatment [208]. Aflibercept is a soluble decoy VEGFR-1 and has approximately hundreds of times the affinity for VEGF-A than bevacizumab [209].

## 6. Conclusions

As the research of MPs and PS has progressed, they have attracted more attention for their roles in various diseases. Externalized PS on MPs derived from tumor cells and blood cells induce the coagulation cascade via the extrinsic and intrinsic coagulation pathways, which in turn regulate the formation of pre-metastatic niches characterized by ECM remodeling, immunosuppression, and angiogenesis. Interestingly, coagulation factors (mainly TF, thrombin, and fibrin) mediate tumor progression and metastasis through various signaling pathways; exploring these effects enriches our knowledge about the interaction of coagulation and tumor metastasis and leads to the development of better therapeutic strategies. The bidirectional association between tumor-induced coagulation activation and tumor growth and metastasis offers the possibility that these processes could be positively influenced by some antithrombotic strategies. Coagulation factor inhibitor-mediated anti-tumor effects include reduced tumor vascularization and matrix remodeling and attenuated adhesion and extravasation of tumor cells to ECs, ultimately ameliorating tumor progression. This review highlights the critical role of PS and its mediated coagulation cascade in tumor metastasis. Targeted PS therapy offers compelling advantages, and the results are encouraging.

## Figures and Tables

**Figure 1 cancers-15-01957-f001:**
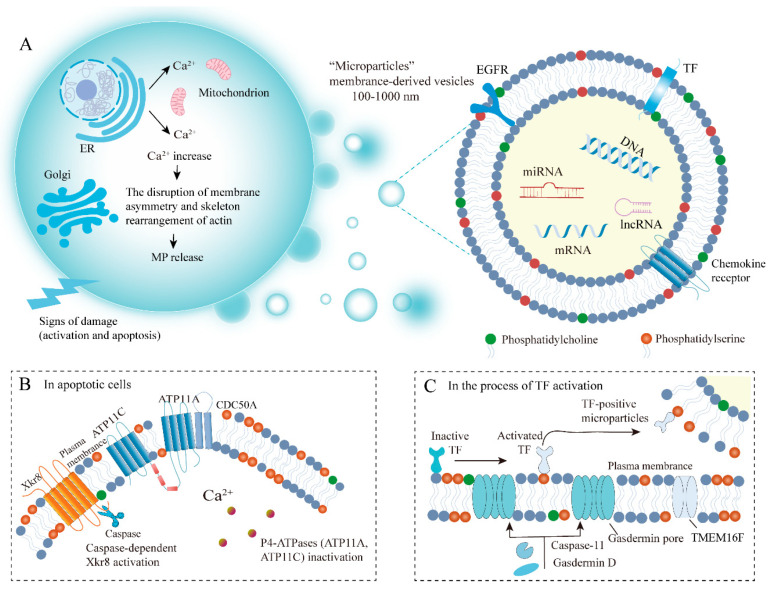
Mechanisms of MP formation and PS externalization. (**A**) Signs of damage (activation and apoptosis) cause increased cytosolic Ca^2+^ concentration, induce membrane asymmetry disruption and actin cytoskeletal rearrangement and promote MP release. (**B**) In cells undergoing apoptosis, caspase-dependent Xkr8 activation and P4-ATPase inactivation together result in persistent PS exposure. (**C**) In the process of TF activation controlled by caspase-11- and gasdermin D-dependent pathways, PS exposure is mediated by TMEM16F. EGFR; Epithelial growth factor receptor; MPs: microparticles; PS: phosphatidylserine; Xkr8: Xk-related protein 8; TF: Tissue factor, TMEM16F: Transmembrane protein 16F.

**Figure 2 cancers-15-01957-f002:**
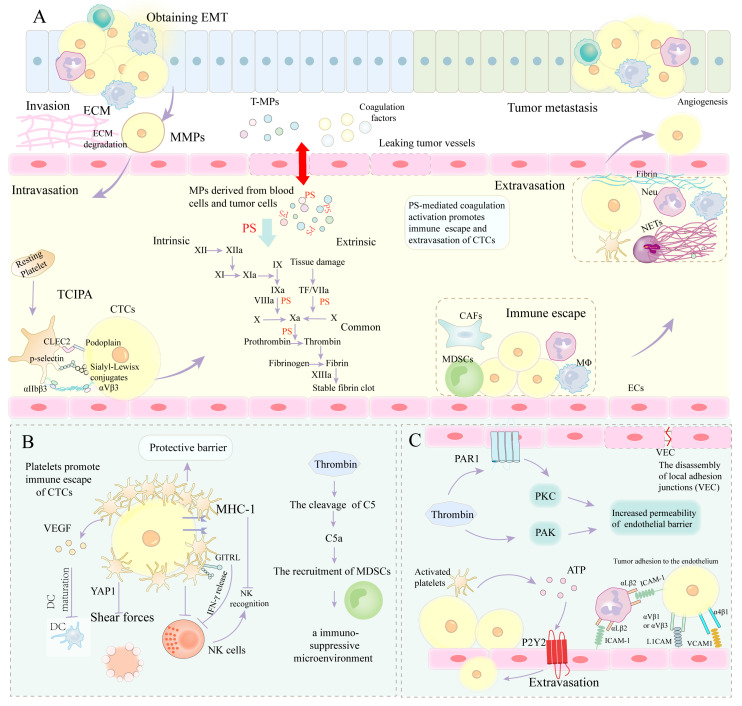
PS-mediated coagulation cascade involved in tumor progression. (**A**) Following platelet activation, platelets tightly bind to tumor cells by various adhesion receptors and promote CTC survival and immune escape. PS on MPs derived from tumor cells and blood cells mediate the coagulation cascade. PS-mediated coagulation cascade protects CTCs from the shear forces of blood flow and immune cell attack. Coagulation activation further promotes the extravasation of CTCs. Coagulation factors enter the tumor microenvironment by leaking tumor vessels and are involved in EMT, ECM degradation, and angiogenesis. (**B**) Platelets promote immune escape of CTCs via multiple mechanisms. Thrombin can directly cleave C5 to generate C5a, which induces an immunosuppressive microenvironment by the recruitment of MDSCs. (**C**) In addition to through integrin α4β1, αVβ1 or αVβ3, tumor cells can adhere to endothelium with the help of leukocytes. Activated platelets release ATP from dense granules, activate endothelial P2Y2 receptors and allow trans-endothelial migration of tumor cells by increasing the permeability of blood vessels. Thrombin activates PKC pathways or mediates PAK activity to increase the permeability of the endothelial barrier. EMT: epithelial-mesenchymal transition; ECM: extracellular matrix; MMPs: matrix metalloproteinases; T-MPs: tumor-derived MPs; TCIPA: tumor cell-induced platelet aggregation; CLEC-2: C-type lectin-like receptor 2, CTCs: circulating tumor cells; PS: Phosphatidylserine; CAFs: cancer-associated fibroblasts; MDSCs: myeloid-derived suppressor cells; mø: Macrophages; Neu: Neutrophils; NETs: Neutrophil extracellular traps, DC: Dendritic cells, YAP1: yes-associated protein-1, GITRL: the glucocorticoid-induced tumor necrosis factor receptor ligand; NK: natural killer, PKC: protein kinase C; PAK; p21-activated kinase, VEC: VE-cadherin, ICAM-1: intercellular adhesion molecule 1, VCAM1: vascular cell adhesion molecule-1,L1CAM: L1cell adhesion molecular; ECs: Endothelial cells; VEGF: Vascular endothelial growth factor.

**Figure 3 cancers-15-01957-f003:**
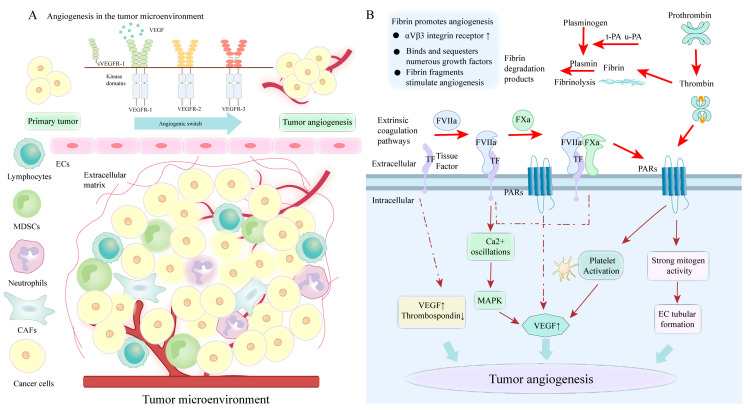
PS-mediated coagulation cascade involved in tumor angiogenesis. (**A**) Angiogenesis in the tumor microenvironment. (**B**) The mechanisms of angiogenesis induced by coagulation factors. TF promotes tumor-directed angiogenesis through upregulating vascular VEGF expression and downregulating the angiogenesis inhibitor (thrombospondin) expression. TF/FVIIa complexes cause Ca^2+^ oscillations and activate MAPK pathway, resulting in VEGF expression. The TF/FVIIa/FXa complexes activate one or more PARs to support angiogenesis. Thrombin/PAR-1 activation can produce strong mitogen activity on ECs and vascular progenitor cells to induce EC tubular formation in the matrix membrane. Additionally, Thrombin/PAR-1 activated platelets lead to their aggregation and degranulation (VEGF). The fibrin matrix can upregulate the expression of αVβ3 integrin receptor to promote angiogenic responses. The fibrin matrix sequesters and protects numerous growth factors (VEGF) from being degraded by proteinase in ECM. Fibrin fragments (E fragments) are shown to stimulate angiogenesis. VEGF: vascular endothelial growth factor; ECs: endothelial cells; MDSCs: myeloid-derived suppressor cells; CAFs: cancer-associated fibroblasts; t-PA: tissue-type plasminogen activator; u-PA: urokinase-type plasminogen activator; TF: tissue factor; PARs: Protease-activated receptors; MAPK: the mitogen-activated protein kinase; FVIIa: activated factor VII; FXa: activated factor X.

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
