# Peer review of "Microparticle Phosphatidylserine Mediates Coagulation: Involvement in Tumor Progression and Metastasis"

_cancers, 2023, doi:10.3390/cancers15071957_

Round 1

Reviewer 1 Report

The role of microparticles and PS externalization on MPs and PS-mediated coagulation cascades in tumor progression and metastasis was expertly described by the authors of this review work. The results of research on cell and animal models are noteworthy. However, some scheme or figure summarizing the PS-mediated coagulation cascade outlined in relation to angiogenesis might be helpful. Overall, the manuscript is well thought out and well organized, and I support the publication of this article.

In addition, I would like to point out some minor corrections related to the language.

  1. Use commas before 'including' (for example, in lanes 127, 532, 729, etc.)
  2. Lane 127: "maintain 'a' dynamic asymmetric homeostasis.." Remove 'a'
  3. Lane 241: PE and DVT, 'have become the second 
  4. Lane 266: correct spelling. 'generation'
  5. Lane 414: remove space 'bloodstream' 
  6. Lane 593: remove space 'molecule 1/2'
  7. Lane 617: blood 'makes' it....

Author Response

Reviewer 1

The role of microparticles and PS externalization on MPs and PS-mediated coagulation cascades in tumor progression and metastasis was expertly described by the authors of this review work. The results of research on cell and animal models are noteworthy. However, some scheme or figure summarizing the PS-mediated coagulation cascade outlined in relation to angiogenesis might be helpful. Overall, the manuscript is well thought out and well organized, and I support the publication of this article.

Response: Thanks for your suggestion. We have added a figure summarizing the PS-mediated coagulation cascade outlined in relation to angiogenesis(Figure 3).

In addition, I would like to point out some minor corrections related to the language.

  1. Use commas before 'including' (for example, in lanes 127, 532, 729, etc.)

Response: Thank you very much for your recommendation. We have added commas before 'including' (in lanes 127, 532, 729, etc.) and marked in yellow.

  1. Lane 127: "maintain 'a' dynamic asymmetric homeostasis." Remove 'a'

Response: Thanks for your suggestion. We have removed 'a' in lane 127.

  1. Lane 241: PE and DVT, 'have become the second

Response: Thanks for your suggestion. We have replaced 'becomes' with ' have become' and marked in yellow.

  1. Lane 266: correct spelling. 'generation'

Response: Thank you very much for your recommendation. We have corrected spelling errors ('generation') and marked in yellow.

  1. Lane 414: remove space 'bloodstream'

Response: Thanks for your suggestion. We have removed space ('bloodstream') and marked in yellow.

  1. Lane 593: remove space 'molecule 1/2'

Response: Thank you very much for your recommendation. We have removed space (' molecule 1/2') and marked in yellow.

  1. Lane 617: blood 'makes' it....

Response: Thank you very much for your recommendation. We have changed 'make' into 'makes' in lane 617 and marked in yellow.

Reviewer 2 Report

It was a nice study about the effects of phosphatidylserine presented on the surface of microparticles on the progression and metastasis of cancer via mediating coagulation. Here are some comments on this study that should be considered before publication:

1.       Please improve the quality of the abstract.

2.       There are some typos- and grammatical mistakes in the text that should be corrected.

3.       Please introduce all the abbreviations at their first-time usage.

4.       “PS Externalization on MPs Involved in Tumor Coagulation” what does coagulation mean here?

5.       Please summarize the text mentioned under heading 4 (lines 341-403).

6.       Please mention the effects of “PS” in each part of section 4.

7.       Section 5.3 is too short, please add more information to this section.

Author Response

Reviewer 2

It was a nice study about the effects of phosphatidylserine presented on the surface of microparticles on the progression and metastasis of cancer via mediating coagulation. Here are some comments on this study that should be considered before publication:

  1. Please improve the quality of the abstract.

Response: Thank you very much for your recommendation. We have improved the quality of the abstract.

  1. There are some typos- and grammatical mistakes in the text that should be corrected.

Response: Thanks for your suggestion. We have corrected typos- and grammatical mistakes in the text.

  1. Please introduce all the abbreviations at their first-time usage.

Response: Thank you very much for your recommendation. We have added all the abbreviations at their first-time usage.

  1. “PS Externalization on MPs Involved in Tumor Coagulation” what does coagulation mean here?

Response: Thanks for your suggestion. Coagulation in this context refers to coagulation activation mediated by multiple factors of the tumor (including PS) leading to a hypercoagulable state or thrombosis. Up to 20% of cancer patients develop vascular thromboembolism, consisting of pulmonary embolism and deep vein thrombosis. Even in the absence of thrombosis, most cancer patients typically present with abnormal laboratory coagulation parameters, and these alterations reveal different degrees of coagulation activation, suggesting a constant subclinical hypercoagulable state.

  1. Please summarize the text mentioned under heading 4 (lines 341-403).

Response: Thank you very much for your recommendation. Firstly, we elaborate on general process of tumor metastasis, including enhanced metastatic properties by obtaining EMT, entry of tumor cells into the circulation through depredating the extracellular matrix by MMPs, immune escape through interacting with blood cells (platelets, etc.), adhesion to endothelial cells and subsequent extravasation by utilizing coagulation and establishing a vascular network.

Secondly, abnormal vascular leakage of tumors allows intravascular platelets and coagulation factors (including thrombin and fibrinogen) to enter the extravascular space and can affect tumor spread and metastasis in the tumor microenvironment.

Finally, coagulation components promote tumor growth and metastatic spread. PS-mediated coagulation activation is involved in multiple processes of tumor progression, including EMT, ECM remodeling, tumor extravasation, and immune escape.

  1. Please mention the effects of “PS” in each part of section 4.

Response: Thanks for your suggestion. Our article first describes the PS-mediated coagulation activation, and then PS indirectly promotes tumor progression through its mediated production of coagulation factors. To fit the overall structure of the article, it is reasonable to present the role of coagulation factors rather than PS in each part of tumor metastasis (including EMT, ECM remodeling, tumor extravasation, and immune escape) in section 4. Additionally, paper give the detail elaboration the mechanism of PS-mediated coagulation activation is given in section 3.

  1. Section 5.3 is too short, please add more information to this section.

Response: Thank you very much for your recommendation. We have added other treatment strategies and marked in yellow.

Targeting integrin therapy

Intetumumab (CNTO 95) is a human anti-αV integrin antibody that inhibits melanoma cell adhesion, migration, and invasion in vitro (Trikha et al, Int. J. Cancer 2004). Abituzimab (DI17E6) is a humanized monoclonal antibody against αV subunit-containing integrins, suppresses migration and invasion of prostate cancer cells in preclinical models (Jiang et al, Mol. Cancer Res. 2017). The orally active αVß3 integrin inhibitor MK-0429 can reduce lung metastasis and melanoma burden in mouse models. MK-0429 is also utilized in patients with hormone-refractory prostate cancer and bone metastases (Rosenthal et al, Asia Pac. J. Clin. Oncol. 2010). D-pinitol modulates FAK, c-Src, and NF-κB pathways to inhibit αVß3 integrin and has anti-metastatic effects in human prostate cancer cells (Lin et al, Int. J. Mol. Sci. 2013). We have marked in yellow.

Trikha, M., Zhou, Z., Nemeth, J.A., Chen, Q., Sharp, C., Emmell, E., Giles-Komar, J., Nakada, M.T. CNTO 95, a fully human monoclonal antibody that inhibits alphav integrins, has antitumor and antiangiogenic activity in vivo. Int. J. Cancer 2004, 110(3), 326-35.

Jiang, Y., Dai, J., Yao, Z., Shelley, G., Keller, E.T. Abituzumab targeting of αV-class integrins inhibits prostate cancer progression. Mol. Cancer Res. 2017, 15(7), 875-883.

Rosenthal, M.A., Davidson, P., Rolland, F., Campone, M., Xue, L., Han, T.H., Mehta, A., Berd, Y., He, W., Lombardi, A. Evaluation of the safety, pharmacokinetics and treatment effects of an alpha(nu) beta(3) integrin inhibitor on bone turnover and disease activity in men with hormone-refractory prostate cancer and bone metastases. Asia Pac. J. Clin. Oncol. 2010, 6(1), 42-8.

Lin, T.H., Tan, T.W., Tsai, T.H., Chen, C.C., Hsieh, T.F., Lee, S.S., Liu, H.H., Chen, W.C., Tang, C.H. D-pinitol inhibits prostate cancer metastasis through inhibition of αVβ3 integrin by modulating FAK, c-Src and NF-κB pathways. Int. J. Mol. Sci. 2013, 14(5), 9790-802.

Targeting HIF therapy

HIFs mediate EMT via multiple signaling pathways, thereby resulting in tumor metastasis and chemoresistance. HIFs are promising therapeutic targets for the management of cancer. Cetuximab reduces HIF-1α mainly at the level of protein synthesis. RX-0047 (a potent inhibitor of HIF-1α) is a 20-mer phosphorothioate antisense oligonucleotide and directly suppresses HIF-1α through reducing the levels of mRNA and protein (Hu et al, J. Cell Biochem. 2013). Non-specific inhibitors of HIF-2α usually inhibit HIF-2α transcription, protein synthesis and nuclear translocation (Yu et al, Yonsei Med. J. 2017). Among the small molecules currently being developed, TC-S 7009, PT-2977 and HIF-2α transcriptional inhibitors are reported as HIF-2α-specific antagonists (Murugesan et al, Drug Discov. Today 2018; Motto et al, J. Chem. Inf. Model 2011). We have marked in yellow.

Hu, Y., Liu, J., Huang, H. Recent agents targeting HIF-1α for cancer therapy. J. Cell Biochem. 2013, 114(3), 498-509.

Yu, T., Tang, B., Sun, X. Development of inhibitors targeting hypoxia-inducible factor 1 and 2 for cancer therapy. Yonsei Med. J. 2017, 58(3), 489-496.

Murugesan, T., Rajajeyabalachandran, G., Kumar, S., Nagaraju, S., Jegatheesan, S.K. Targeting HIF-2α as therapy for advanced cancers. Drug Discov. Today. 2018, 23(7), 1444-1451.

Motto, I., Bordogna, A., Soshilov, A.A., Denison, M.S., Bonati, L. New aryl hydrocarbon receptor homology model targeted to improve docking reliability. J. Chem. Inf. Model 2011, 51(11), 2868-81.

Targeting VEGF therapy

Thrombin promotes tumor angiogenesis mainly through VEGF, which makes it an attractive target for anti-angiogenic therapy (Ferrara et al, Oncologist. 2004). VEGF inhibitors block the activation of VEGFR receptors (Wedam et al, J. Clin. Oncol. 2006) and further intracellular signaling, thereby inhibiting endothelial cell proliferation and migration, and preventing the formation of tumor vascular networks. Following FDA approval, bevacizumab is widely used in clinical practice in the United States. In glioblastoma diagnosis, one-third of patients have no tumor progression within 6 months following treatment (Chekhonin et al, Curr. Cancer Drug Targets 2013). Aflibercept is a soluble decoy VEGFR-1 and has approximately hundreds of times the affinity for VEGF-A than bevacizumab (Wachsberger et al, Int. J. Radiat. Oncol. Biol. Phys. 2007). We have marked in yellow.

Ferrara, N. Vascular endothelial growth factor as a target for anticancer therapy. Oncologist. 2004, 9 Suppl 1, 2-10.

Wedam, S.B., Low, J.A., Yang, S.X., Chow, C.K., Choyke, P., Danforth, D., Hewitt, S.M., Berman, A., Steinberg, S.M., Liewehr, D.J., et al. Antiangiogenic and antitumor effects of bevacizumab in patients with inflammatory and locally advanced breast cancer. J. Clin. Oncol. 2006, 24(5), 769-77.

Chekhonin, V.P., Shein, S.A., Korchagina, A.A., Gurina, O.I. VEGF in tumor progression and targeted therapy. Curr. Cancer Drug Targets 2013, 13(4), 423-43.

Wachsberger, P.R., Burd, R., Cardi, C., Thakur, M., Daskalakis, C., Holash, J., Yancopoulos, G.D., Dicker, A.P. VEGF trap in combination with radiotherapy improves tumor control in u87 glioblastoma. Int. J. Radiat. Oncol. Biol. Phys. 2007, 67(5), 1526-37.

The content marked in yellow in the manuscript are responses to reviewer comments.

Round 2

Reviewer 2 Report

Thank you from the authors to revise the manuscript, the following comment remains that should be corrected:

1- Section 4 should be summarized in the main text.